# Changes in the Fruit Quality Parameters of Medlar Fruit (*Mespilus germanica* L.) after Heat Treatment, Storage, Freezing or Hoarfrost

**DOI:** 10.3390/foods12163077

**Published:** 2023-08-16

**Authors:** Maja Mikulic-Petkovsek, Katja Jakljevic, Robert Veberic, Metka Hudina, Denis Rusjan

**Affiliations:** Department of Agronomy, Biotechnical Faculty, University of Ljubljana, Jamnikarjeva 101, SI-1000 Ljubljana, Slovenia; jakljevi.katja@gmail.com (K.J.); robert.veberic@bf.uni-lj.si (R.V.); metka.hudina@bf.uni-lj.si (M.H.); denis.rusjan@bf.uni-lj.si (D.R.)

**Keywords:** medlar fruit, fruit treatment, storage, temperature, HPLC, phenolic compounds, sugars, acids

## Abstract

The present study deals with the comparison of traditional fruit processing methods on medlar fruits and their effects on sugar content, organic acids, and phenolic composition in the medlar fruit variety ‘Domača nešplja’. The study aimed to analyze which processing methods can be used to make technologically mature medlar fruits that are not yet suitable for consumption edible and to maintain their good chemical quality. The two major sugars in medlars are fructose (59.30 g/kg FW) and glucose (54.43 g/kg FW), and the most abundant organic acids present are malic (8.44 g/kg FW) and quinic acid (8.77 g/kg FW). A total of 38 different phenolic compounds were identified in the medlar fruits: 13 phenolic acids, 9 flavanols, 1 flavone, 3 flavanones, and 12 flavonol glycosides. To explicate: phenolic acids (532.85 mg/kg FW) and flavanols (375.21 mg/kg FW) predominated; neochlorogenic acid had the highest content among phenolic acids; and procyanidins were the most abundant flavanols. The analysis observed statistical differences in metabolite content amongst fruits treated differently (technologically ripe fruits (harvested from the three fruits), edible fruits (technologically ripe fruits stored at 8 °C for 25 days), fruits exposed to the hoarfrost (temperature −1 °C to −4 °C), fruits heated at 60 °C (3 h), and frozen fruits (at −20 °C for 2 months). The lowest levels of fructose (191.77–195.1 g/kg DW) and sorbitol (29.35–31.3 g/kg DW) were detected in the heated and edible fruits. Edible fruits had a 30% lower content of organic acids than technologically ripe fruits and a five times lower content of flavanols, whereas flavonols had an 18.7 times lower content of phenolic acids than technologically ripe fruits. Heating the fruits to 60 °C resulted in a 40% increase in total phenolic compounds in medlars. The results of the study indicate that exposure of medlar fruit to hoarfrost does not significantly affect the chemical quality of the fruit and only minimally alters the composition of sugars, acids, and phenolic compounds. The processing of medlar fruit with hoarfrost, therefore, remains the most suitable method of fruit bletting.

## 1. Introduction

The common medlar (*Mespilus germanica* L.) originates in the Caspian Sea area. Throughout history, medlar has decorated monasteries’ courtyards and gardens, been planted in parks and green areas, and been depicted in the works of various artists. Nowadays, the species is mainly distributed in Turkey, Iran, and certain parts of Europe [1,2]. Medlar is grown in the form of a shrub or tree [3]. It can be grafted onto seedlings of medlar (*Mespilus germanica* L.), hawthorn (*Crataegus laevigata* Poir), pear (*Pyrus communis* L.), and quince (*Cydonia oblonga* Mill.) [1,4]. Among the genotypes of medlar, there is great variability in the shape, color, and taste of the fruit and the content of chemical components [5]. Medlar is a typical climacteric fruit. The fruit contains a stony seed; its skin is light to dark brown; and its flower normally has a wide sepal [6].

In the literature, medlar fruit is considered to have high nutritional value because it contains sugars, organic acids, dietary fiber, carotenoids, and minerals such as phosphorus, calcium, magnesium, sodium, and potassium [2,7,8]. It contains a high amount of fructose and a lower amount of xylose, glucose, and sucrose, whereas the content of organic acids varies greatly among genotypes [2].

Medlar fruits are rich in bioactive compounds, are a good source of natural antioxidants, and are consequently often used in traditional folk medicine to relieve various problems [3,4,9,10,11]. To illustrate, numerous studies have reported that phenolic compounds are responsible for the curative effects of different fruit species since they exert a positive effect on reducing chronic disease incidence [6,12,13]. Medlar fruits contain various phenolic constituents, of which tannins, flavonoids, and phenolic acids stand out in their quantity [14,15,16]. It is known that the content of bioactive substances in fruits depends on the type of fruit, cultivar, environmental conditions (temperature, precipitation, soil, etc.), growing period, stage of fruit ripening [17], and also on the technology of fruit processing [18,19,20,21]. Medlar fruits are used for fresh consumption and various alcoholic beverages; they are processed into jam, marmalade, puree, pickles, or candied [22].

From a nutritional point of view, it is usually best to eat fruits fresh because they contain the most valuable vitamins and other ingredients. However, this is not true for medlar because its fruits are very firm and have an astringent taste when they are technologically ripe. This is the reason why it is not suitable for consumption and needs to be processed in some way [5]. Various fruit processing methods are, therefore, used to process the fruit and prolong its availability to the consumer throughout the year (e.g., heat treatment, drying, freezing, pickling, ultrasound, etc.) [23,24].

However, all processing techniques indeed affect the microstructure of the fruit, its color, flavor, and nutritional composition. Usually, after fruit processing, the quality parameters of the fruit are reduced (e.g., reduction of nutrient content, certain bioactive compounds, as well as sensory properties). In general, it is always better to use methods of fruit processing that do not drastically deteriorate the quality of the fruit or its product in order to retain the taste and health benefits [25,26].

Medlar fruits are normally not eaten fresh but should be left on the tree for a while or stored in a cellar until they become soft and edible. This process is called “bletting” and can also be accelerated by hoarfrost [1,27]. During the softening of medlar fruits, the fruits become darker brown due to the activity of polyphenol oxidase [28], and the starch in the fruit is inverted to sugar, which signifies that the sugar content increases while the tannin content decreases [4]. The taste of medlar fruits is quite specific and is reminiscent of a combination of pear and apple.

What is most challenging with medlars is achieving a good edible quality of the fruit. When technically ripe fruits are picked from the tree, they are unsuitable for consumption because they are very firm and have a bitter taste. Medlar fruits are, therefore, suitable for consumption only when bletted. From a practical and scientific point of view, consumers and food technologists ask which process makes medlars suitable for consumption without too much decrease in the content of chemical substances in the fruit. It appears that no research has been conducted on this topic with medlars and there are still many unanswered questions in this area. It was thus decided to research how different fruit processing methods affect the chemical composition of medlar fruit. The main interest was to discover whether the content of sugars, organic acids, and phenolic compounds in medlar fruit changes significantly with different processing methods. The analysis tried to observe which processing method would give the best chemical quality of the fruit, acknowledging that medlar fruits of the local genotype ‘Domača nešplja’ have not been chemically analyzed before. This study’s results should provide original and useful information for the scientific community, the food industry, private institutions, and consumers.

## 2. Materials and Methods

### 2.1. Plant Material

Fruits of common medlar (*Mespilus germanica* L.) ‘Domača nešplja’ were harvested on 20th November 2020 in Dr. Derganca’s learning garden in Semič (lat 45°39′06′′, long 15°10′44′′, 246.62 m MSL). The trees of common medlar were 12 years old and planted at a 7 × 8 m planting distance. At harvest, the fruits were technologically ripe. Medlar fruits were treated according to five different processing methods. Each processing method included 60 fruits, and for each treatment, six replicates were performed.

### 2.2. Fruit Treatments

The following processing methods were used: (1) technological maturity—technologically ripe fruits picked directly from the trees (fruits were immediately extracted and analyzed), (2) deep freezing—frozen technologically ripe fruits at −20 °C for two months, (3) edible maturity—edible ripe fruits obtained by storing technologically ripe fruits in a dark cellar at a temperature of 8 °C for 25 days, (4) heating—technologically heated ripe fruits for 3 h at a temperature of 60 °C, (5) hoarfrost—edible ripe fruits harvested directly from the tree after the fruits were exposed to three hoarfrosts (minimum temperatures from −1 °C to −4 °C, harvested on 3 December 2020). Medlars from each treatment were extracted for the analysis of sugars, organic acids, and phenolic compounds.

### 2.3. Extraction and Analysis of Sugars and Organic Acids

Extracts of sugars and acids from medlar fruits were prepared using the method cited by Mikulic-Petkovsek et al. [29]. Six repetitions were performed for each treatment. The fruits were cut into small pieces and further homogenized in a mortar. 5 g of the material was weighed into centrifuges and poured into 20 mL of double-distilled water. The prepared samples were shaken on a shaker for half an hour to completely extract all sugars and acids from the fruits. The homogenate was centrifuged for 8 min at 10 °C at 15.695× *g* (Eppendorf Centrifuge 5819 R, Eppendorf, Hamburg, Germany). The supernatant was filtered through a 0.20 μm cellulose mixed ester filter (Macherey-Nagel, Düren, Germany) into a glass vial. Sugars and acids were analyzed by Thermo Fischer high-performance liquid chromatography (HPLC) (Vanquish ^TM^ Flex UHPLC). An RI detector and a Rezex RCM monosaccharide Ca + column (2%) (300 × 7.8 mm, Phenomenex) at an operating temperature of 80 °C were used to identify sugars. For the identification of organic acids, a UV detector at 210 nm and a Rezex ROA—organic acid H + column (8%) (300 × 7.8 mm, Phenomenex) heated to 65 °C were used. The mobile phase used for the analysis of sugars was bi-distilled water, and for organic acids, sulfuric acid in water. In both analyses, the flow of the mobile phase was 0.6 mL/min. Calibration curves were prepared using standard solutions of sucrose, glucose, sorbitol, fructose, citric acid, malic acid, quinic acid, fumaric acid, and shikimic acid. Deionized water was used to prepare the sugar and acid standards. Each standard was prepared at four different concentrations, and the content of individual sugars and organic acids was calculated from the calibration curve of the corresponding standards. The contents were expressed in g/kg or mg/kg of fresh (FW) or dry (DW) medlar fruit.

### 2.4. Determination of Fruit Dry Weight and Conversion of Metabolites to Fresh or Dry Weight

Since the analysis was interested in the content of the analyzed metabolites in fresh fruits, all the contents on a fresh weight basis were expressed in the first part of the study (FW). The results can, therefore, also be compared with the literature. In the case of differently treated medlars (frozen, heated, technologically ripe, harvested after hoarfrost, and stored in a cellar), the results were expressed in dry weight (DW) because the fruit with different treatments contained varying proportions of water. To determine the percentage of dry weight, fresh medlar fruits from each treatment were dried at a temperature of 105 °C for 72 h until a constant mass was achieved. The percentage of dry matter in the fruit was obtained from the difference between the masses before and after drying.

### 2.5. Extraction and Analysis of Phenolic Compounds

The extraction of phenolic compounds was performed according to the method described by Mikulic-Petkovsek et al. [29]. There were six repetitions per treatment. For the extraction of phenolic compounds, 5 g of chopped medlar fruits were weighed and further crushed in a mortar. The samples were then poured over with 8.5 mL of 80% methanol (80%/20% = methanol/water), shaken, and transferred to an ultrasonic bath for one hour, to which ice was added to keep the extraction at a low temperature, i.e., around 2–3 °C. The ultrasound-assisted extraction was conducted at a frequency of 30 kHz and an input power of 400 W. The homogenate was then centrifuged (Eppendorf Centrifuge 5810 R) at 5 °C for 8 min at 10,000 rpm. The supernatant was filtered through polyamide/nylon filters (Macherey-Nagel, Düren, Germany) into vials and numbered appropriately. The analysis of individual phenolic compounds was performed by HPLC (Thermo Scientific Dionex, Waltham, MA, USA) with a DAD detector at two wavelengths (280 nm and 350) using a Gemini C18 column (Phenomenex, Torrance, CA, USA). Mobile phase A was a combination of 0.1% formic acid/3% acetonitrile/96.9% bi-distilled water, and mobile phase B was 0.1% formic acid/3% water/96.9% acetonitrile. Mobile phase elution was achieved according to a linear gradient from 5 to 20% B in the first 15 min, followed by a linear gradient from 20 to 30% B for 5 min, then an isocratic mixture for 5 min, followed by a linear gradient from 30 to 90% B for 5 min, and then an isocratic mixture for 15 min before returning to the initial conditions [30]. The mobile phase flow was 0.6 mL min^−1^, and the volume of extract injection was 20 μL. Identification of phenolic compounds was performed with a mass spectrometer (LTQ XL Linear Ion Trap Mass Spectrometer, Thermo Fisher Scientific, Waltham, MA, USA) with electrospray negative ionization. The analyses were carried out using full-scan data-dependent MSn scanning from *m/z* 115 to 1900. The source parameters were: capillary temperature of 250 °C; sheath gas and auxiliary gas of 60 and 15 units; source voltage of 3 kV; and normalized collision energy of 20% to 35%. Spectral data were elaborated using Thermo Scientific™ Xcalibur™ 4.3 software. Phenolic compounds were identified based on their retention times and PDA spectra compared with phenolic standards and fragmentation patterns compared to literature data. The content of individual phenolic compounds was calculated using standard curves for different phenolics. The following standard curves were used: protocatechuic acid (Y = 563.5×), 3-caffeoylquinic acid (Y = 510.09×), 5-caffeoylquinic acid (Y = 466.48×), caffeic acid (Y = 1051.5×), sinapic acid (Y = 507.11×), procyanidin B1 (Y = 74.34×), *p*-coumaric acid (Y = 1742.52×), ferulic acid (Y = 1246.8×), catechin (Y = 116.26×), ellagic acid (Y = 150.06×), epicatechin (Y = 185.65×), apigenin-7-glucoside (Y = 549.68×), naringenin (Y = 351.12×), quercetin-3-galactoside (Y = 656.53×), quercetin-3-glucoside (Y = 684.67), quercetin-3-rutinoside (Y = 436.53×), myricetin-3-glucoside (642.17×), kaemperol-3-glucoside (Y = 585.16×), isorhamnetin-3-glucoside (529.29×). Since no standards were available for some phenolic compounds, their contents were given as equivalents of related substances. Thus, quercetin pentoside, quercetin dirhamnoside, and quercetin rhamnosyl hexoside were reported as quercetin-3-galactoside; dihydrokaempferol hexoside; kaempferol hexoside 1 and 2 as kaempferol-3-glucoside; laricitrin rhamnoside as myricetin-3-glucoside; isorhamnetin hexoside 1 and 2 as isorhamnetin-3-glucoside; apigenin hydroxyhexoside as apigenin-7-glucoside; all naringenin derivatives as naringenin; 3-*p*-coumaroylquinic acid, 4-*p*-coumaroylquinic acid, and 5-*p*-coumaroylquinic acid 1 and 2 as *p*-coumaric acid; 3-feruloylquinic acid 1 and 2 as ferulic acid; caffeic acid hexoside as caffeic acid; and all procyanidin derivatives as procyanidin B1. The content of phenolics was expressed in mg/kg FW or mg/kg DW. The content of all identified individual phenolics was summed and presented as the total analyzed phenolic content.

### 2.6. Statistical Analysis

Statistical analysis was performed using the R-commander program. The analysis was performed with a one-way analysis of variance (ANOVA). The levels of sugars, organic acids, and phenolic compounds were compared among the different treatments. Statistically significant differences between treatments were compared using the Duncan test at a 95% confidence level. The contents of metabolites in differently treated medlars were expressed per dry weight (DW), and the content in technologically mature medlars was expressed per fresh weight (FW). The results for technologically mature medlar were recalculated in Microsoft Excel 2010. Principal component analysis (PCA) was performed for the content of individual sugars, organic acids, and all phenolic compounds, as well as for phenolic groups (flavanols, flavonols, phenolic acids, flavones, and flavanones) in the variously treated medlar fruits. PCA analysis was performed on standardized variables. Cluster analysis was performed using the Euclidean distance coefficient using the Ward method using the R-commander program.

## 3. Results and Discussion

### 3.1. Content of Sugars, Acids and Phenolic Compounds in Fresh Technologically Mature Medlar Fruits

Glucose and fructose were the predominant sugars in medlar fruits, accounting for almost 90% of the total sugar content (127.16 g/kg FW) (Table 1). Compared to other pome fruits, medlars have lower sugar content than apples (128.2 to 191.6 g total sugar/kg) [31] but higher than pears (52.3 to 99.9 g total sugar/kg FW) [32]. Sucrose (4.07 g/kg FW) and sorbitol (9.46 g/kg FW) were also found in medlar fruits. The latter is a typical sugar for certain fruit species and can also be used as a distinguishing characteristic for the authenticity of pure fruit juice. The results indicated that the content of glucose (54.43 g/kg FW) and fructose (59.20 g/kg FW) in the medlar ‘Domača nešplja’ is slightly higher than in other medlar genotypes.

Previously reported values for fructose in medlar ranged from 32.5 to 47.26 g/kg FW and from 21.08 to 30.17 g/kg FW for glucose [2]. Due to the high glucose and fructose content, ‘Domača nešplja’ has a high total sugar content, which is 1.5 to 2.2 times higher than the eleven Turkish medlar genotypes studied, which ranged from 58.29 to 84.96 g total sugar/kg FW [33].

In fruit species, malic acid and citric acid are usually the predominant acids [34], while in medlar, quinic acid and malic acid prevail, accounting for 94% of the total acids (Table 1). In this study, technologically ripe fruits contained 8.77 g quinic acid/kg FW and 8.44 g malic acid/kg FW. The content of citric acid was only 1 g/kg FW. Fumaric acid and shikimic acid were present in trace amounts and accounted for only 40 mg/kg fruit altogether, which is comparable to the results of Cosmulescu et al. [35], who reported fumaric acid contents in medlar fruits ranging from 1.5 to 12.8 mg/kg FW. This study’s values of malic and citric acid are in the same range as those of Cevahir and Bostan [2] and Ozturk et al. [36] and slightly higher than those of Glew et al. [37].

The taste of the fruit depends on the ratio between the content of sugars and acids (S/A ratio = total sugars/total acids), which suggests that fruits with a high total sugar content do not necessarily taste sweet. In medlar, the S/A ratio is about 7 (Table 1), meaning that the fruit has a slightly sweet to slightly sour taste. Some fruit species have a very low sugar/acid ratio, e.g., black currant, red currant, and cranberry (S/A ratio around 3–4), which indicates a sour taste, whereas some have a high S/A ratio, e.g., eastern shadbush, black mulberry, and goji berry (S/A ratio above 10), which indicates a sweeter taste [38].

Medlar fruits have a very broad spectrum of polyphenolic substances. We found 13 phenolic acids, 9 flavanols, 3 flavanones, 1 flavone, and 12 flavonols (Appendix A, Figure 1). The group of phenolic acids is the most important since it accounts for 57% of the total phenolic content (TAP), i.e., 532.85 mg of phenolic acids per kg FW. Among them, neochlorogenic acid stands out with a content of 473.74 mg/kg FW. Chlorogenic acid (19.81 mg/kg FW) and cryptochlorogenic acid (11.3 mg/kg FW), pertaining to the group of phenolic acids, were also identified, as were four different coumaroylquinic acids, two feruloylquinic acids, sinapic acid, protocatechuic acid, and ellagic acid. Other authors [9,10,15] have also detected ferulic acid, *p*-coumaric acid, caffeic acid, and gallic acid in medlar.

In medlar fruit, flavanols account for 40% of the total phenolic content (375.21 mg/kg FW); this analysis detected the following ones: epicatechin, catechin, four procyanidin trimers, one dimer, and two procyanidin tetramers (Appendix A). The content of catechin in this study’s medlar fruits was similar to the one reported by Rop et al. [15], while the epicatechin content in the genotype ‘Domača nešplja’ was slightly higher. Procyanidin tetramers (144.97 mg/kg FW) and procyanidin trimers (105.88 mg/kg FW) predominated in quantity. In comparison, Gulcin et al. [1] did not identify any flavanols in an aqueous extract of medlar fruit. Flavanols, especially procyanidins and tannins, have been reported to contribute to the bitter and astringent taste of plant foods [39,40]. However, in medlar fruit, the high content of neochlorogenic acid (pseudotannin) and procyanidin tri- and tetramers mostly contributes to it. Pseudotannins have similar properties to tannins; a high content of pseudotannins also signifies that the food has an astringent taste. Examples of pseudotannins include chlorogenic acid in coffee and catechins in cocoa [40].

Only three naringenin hexosides were identified from the flavanones group, and only one representative (apigenin hydroxyhexoside) of flavones was identified. The content of both phenolic groups in medlar fruit was less than 4 mg/kg FW. The group of flavonols was quite varied, as six quercetin derivatives, three kaempferol glycosides, two isorhamnetin derivatives, and one laricitrin derivative were detected in medlar fruit. Kaempferol-3-glucoside had the highest content of the group of flavonols: 11.44 mg/kg FW, while the other representatives had a content of less than 1 mg/kg FW. The total content of the analyzed flavonol glycosides was 16.84 mg/kg FW of medlar, which was less than 2% of the total analyzed phenolic content (TAP). A review of the previous literature revealed that a detailed analysis of the phenolic profile of medlar fruit has not yet been carried out; therefore, the diversity of phenolics analyzed thus far is very limited. Other authors have also reported the presence of quercetin-3-ramnoside [15] and quercetin [15,36], but these two compounds were not confirmed by mass spectrometry.

### 3.2. Changes in Sugar, Organic Acid and Phenolic Content of Medlar Fruit Treated with Different Processing Methods

The most considerate changes in the content of individual sugars during the treatments were observed for fructose and glucose (Table 2). Heating and storing fruits in the cellar resulted in a decrease in fructose and sorbitol content. However, when glucose changes were analyzed, it was found that the frozen fruits had the highest glucose content (204.38 g/kg), while the medlar fruits stored in the cellar contained significantly less glucose (176.92 g/kg DW). No significant changes were observed in sucrose content between the treatments studied (Table 2). These changes in individual sugars also affected the total sugar content in medlars. Thus, the technologically ripe fruits and the frozen fruits had significantly the highest total sugar contents, while total sugar content decreased by 10% in the fruits stored in the cellar and by 6% in the heated fruits compared to the technologically ripe fruits. Sharma et al. [19] also reported that heating fruits engendered a decrease in sugar content; they found that increasing the heating temperature of onions resulted in a decrease in the content of all the sugars analyzed [19]. The reason for the decrease in sugar content after heat treatment is primarily due to the conversion of simple sugars into organic acids, which is a result of tautomerization and retro-aldol reactions, as Jang et al. [41] explain.

Storage time certainly had a major effect on the change in sugar content, which was drastically reduced in medlars during the postharvest period (Table 2). The reason for the decrease in sugar content in medlar fruits during storage is the loss of monosaccharide sugars due to fruit respiration. Since medlar fruits contain low concentrations of sucrose, their hydrolysis into glucose and fructose is also hindered. As a result, the total sugar content of medlars decreases during storage in an uncontrolled atmosphere. Similar findings were also made by Glew et al. [33]. In contrast, Selcuk and Erkan [42] reported that the fructose and glucose content in medlar fruits increased during the first 15–30 days of storage and then decreased compared to the initial sugar content.

The results show that the highest content of malic acid as well as total acids is found in the fruits harvested after hoarfrost or after heating at 60 °C (Table 3). It is, therefore, concluded that neither a temperature around 60 °C nor a low temperature (around 0 °C) causes the degradation of acids. Similar results are reported for pears, where heating fruits for two or three days at 38 °C causes a significant increase in organic acids [43]. On the contrary, heat treatment of poncan fruit accelerates the reduction of tartaric and citric acid content. In fruits stored in an ordinary cellar for 25 days, significantly lower levels of quinic acid, malic acid, shikimic acid, and total acid were observed (Table 3). Ozturk et al. [36] found that malic acid content in medlars decreased by slightly more than 50% after 60 days of storage. In comparison, the fruits from the cellar in this study had a 27% lower total organic acid content than the technologically mature fruits. Glew et al. [37] also reported that after four weeks in storage, medlar fruits of the cultivar ‘Dutch’ had a statistically lower organic acid content than at the beginning of storage. The decrease in the content of organic acids during storage reflects the fact that the organic acids serve as a substrate for climacteric fruit respiration [44]. Some climacteric fruit species intensively use malate as a substrate during the intensive respiration phase [45], while other fruit species, such as papaya, use citrate more rapidly in respiration [46]. Based on this study’s results, it appears that the content of primary metabolites, both sugars and organic acids, statistically decreases during prolonged storage in an ordinary cellar. Hence, it is better to leave medlars on the tree and harvest them after they have been exposed to hoarfrost a few times because fruits retain sufficient content of citric, quinic, and malic acids, as well as total acids and sugars. According to the findings of Selcuk and Erkan [42], it is best to store fruits in a controlled atmosphere if possible. For medlars, these conditions are 2% O_2_ and 5% CO_2_, since malic and citric acids are best retained under these conditions.

The results showed that most phenolic compounds in medlar fruits decreased during storage in the cellar (Table 4). The fruits stored in the cellar thus had a 19 times lower content of phenolic acid (102.28 mg/kg DW) compared to the other treatments. The largest decrease was observed in neochlorogenic acid, which is a major phenolic acid in fruits. Its content in the stored medlar fruits decreased 24.7 times in comparison with the technologically ripe fruits. Similar findings, i.e., that phenolic acids decreased during storage time in medlar fruits, were also reported by Ozturk et al. [47]; in contrast, the variation pattern of phenolic acids was different during storage of pomegranate fruits. The content of some phenolic acids decreased while others increased [48]. In contrast, edible heated medlars from the cellar had significantly the highest levels of 4-*p*-coumaroylquinic acid and 5-*p*-coumaroylquinic acid 1 and 2 (Table 4).

The heated fruits contained the highest content of flavanols (catechins, epicatechins, and procyanidins) and total flavanols (2575.49 mg/kg DW) (Table 4), followed by the fruits treated with hoarfrost and the frozen fruits, which showed a 1.7 to 1.9 lower content of total flavanols compared to the heated fruits. The higher content of flavanols in the heated medlars can be accounted for by the fact that a temperature of 60 °C causes a blockage of the enzyme polyphenol oxidase, which is crucial for the degradation of phenolic substances. Moreover, an increase in flavanols, especially gallocatechin and gallocatechin gallate, was also observed in green tea extracts after heat treatment [49]. The lowest content of flavanols was found in the fruits stored in the cellar for 25 days; they contained only 283 mg flavanols/kg DW. The obtained results differ from the expected ones because, at the beginning of the analysis, it was assumed that the technologically ripe fruits would have the highest flavanol content, while the medlars collected after hoarfrost and the heated medlars would have a lower flavanol content. It is known that medlar fruits lose their astringency when repeatedly exposed to hoarfrost. As expected, the lowest flavanol levels were found in the fruits from the cellar since flavanols are known to degrade during storage. Zhang et al. [50] reported that flavanols are the major phenolic components in lychee fruits, and their levels decreased dramatically during storage. They suggested that flavanol monomers and dimers are more often substrates for enzymatic oxidation than simple phenolic molecules.

Among flavonols, the technologically ripe medlars were characterized by the highest content of laricitrin-3-rhamnoside (2.90 mg/kg DW), while the significantly highest content of isorhamnetin glycosides was found in the heated medlars (2.72 mg/kg DW). The highest levels of quercetin and kaempferol derivatives were found in the technologically ripened medlars, followed by the heated, frozen, and hoarfrost-treated medlars. The significantly lowest flavonol contents were determined in the medlars stored in cellars (12.15 mg total flavonols/kg DW).

Heating the fruits contributed to an increase in total flavones in the fruit samples (14.61 mg/kg DW), in contrast to the stored fruits from the cellar, which had the lowest content of flavones (0.85 mg/kg FW) (Table 4). The results of the measurement of total flavanones were similar, as the fruits from the cellar also had the lowest content of flavanones (0.86 mg/kg FW). Previous publications have also reported that the ripening process leads to a decrease in the content of polyphenolic compounds in medlar fruits [15], which is associated with increased activity of the enzyme polyphenol oxidase [51]. Ozturk et al. [36] found that the decrease in flavonoid content is very pronounced during the storage period and can be easily slowed down by certain storage techniques, e.g., modified atmosphere packaging (MAP), or by treating medlars with certain agents, e.g., methyl jasmonate (MeJA).

Considering the response of each treatment on the content of total phenolics analyzed (TAP), it can be observed that heating medlars causes a 40% increase in phenolics compared to technologically ripened fruits (Table 4). Previous studies suggest that the temperature range of heat treatment is also crucial for changes in phenolic content. For example, in mulberry juice, a temperature between 25 and 45 °C resulted in a decrease in flavonoid content, while a temperature between 45 and 100 °C resulted in an increase in flavonoid content [52]. Opposite results were obtained by Sharma et al. [19], who conducted an experiment with onions and found that heating samples to 120 °C resulted in increased flavonoid content, while a temperature of 150 °C caused its decrease. The effect of heat treatment on the content of certain phenolic substances does not depend only on the temperature but also on the treated food matrix. To support this with the analysis of buckwheat, when buckwheat bran was treated at 180 °C for half an hour, the content of quercetin derivatives decreased, while their content at the same conditions in buckwheat flour increased [53]. Choi et al. [18] also found that both polyphenol content and antioxidant activity increased with longer heating times and higher temperatures. The reason for the better flavonoid extraction could be the conversion or release of units bound to the aglycone or the conversion of complex phenolics into free phenolic glycosides. Another possible reason is that heating the fruit to 60 °C blocks the action of the enzyme polyphenol oxidase and consequently causes better or more efficient extraction of the polyphenolic substances but not their degradation. Similar results were reported for eggplant [54] and lentils [55], where the highest activity of the polyphenol oxidase was at temperatures up to about 35 °C and decreased dramatically at higher temperatures. In contrast, storing fruits in an uncontrolled atmosphere resulted in a significant decrease in total phenolics in medlar fruits. However, there were no significant differences in total phenolic content between technologically ripe fruits (3346 mg/kg DW), fruits harvested after three hoarfrosts (3034 mg/kg DW), or frozen fruits (3477 mg/kg DW). Similar results were also obtained by Ozturk et al. [36], who found that the content of total phenolics in medlars decreased significantly during storage. This can be attributed to the increased activity of polyphenol oxidases [56]. However, in order to preserve the phenolic components in the fruits, their activity can be slowed down by an increased CO_2_ concentration and a decreased oxygen concentration in the storage chamber [42].

The PCA analysis was performed for the content of primary metabolites (sugars and organic acids) and phenolic compounds (flavanols, flavonols, flavanones, flavones, and phenolic acids) in medlars treated in different ways (technological maturity, edible maturity, hoarfrost, heating, and deep freezing). The PCA biplot shows the results of the PC samples (dots) and the loading of the variables, i.e., “loading plots” (blue arrows) (Figure 2). The first component of the full data set of the PCA model explained 69.84% of the variability, and the second component explained 17.19%. It can be observed that two groups are evidently different from the others: the first group contains medlar fruits at edible maturity, which have a significantly lower content of sugars, organic acids, and phenolic components compared to the other treatments, while the second group contains heated medlar fruits, which have a higher content of flavanols and flavones and a lower content of other metabolites.

During fruit storage, numerous changes occur because of physiological processes in the fruit, such as fruit respiration and the action of enzymes, including browning, softening, and a reduction in the nutritional and organoleptic quality of the fruit. From the results of previous studies, minor losses of nutrient content occur when fruits are stored under conditions of elevated CO_2_ concentration, and major damage may also occur in this case due to the enzymatic browning of fruits [42].

## 4. Conclusions

The results of this study, therefore, suggest that technologically mature and ripened medlar fruits have a higher content of individual and total phenolics than bletted medlar fruits stored in an ordinary cellar for 25 days. However, technologically mature medlars do not taste good because they have a very astringent flavor. The main phenolic groups in medlars are phenolic acids (57% of TAP) and flavanols (40% of TAP). In medlars stored in cellars, the content of organic acids, flavones, flavanols, phenolic acids, and, consequently, total phenolics is significantly reduced, even by more than 85% compared to technologically mature medlars. Heat treatment resulted in an increase in total phenolics in the fruits, meaning that a temperature of 60 °C did not affect their degradation, and the extraction obtained was even better. The results of the study show that the edible ripe medlar fruits harvested after triple exposure to hoarfrost had a slightly lower phenolic content than the technologically ripe fruits but were still large enough to have a beneficial effect on human health. Hence, it is believed that edible ripe medlar fruits harvested after hoarfrost have good chemical quality and that this is still the most ideal way to store medlar fruits. In the future, the results of this research could be improved by additional treatments, such as preserving medlar fruits with different storage techniques and processing them at different temperatures.

## Figures and Tables

**Figure 1 foods-12-03077-f001:**
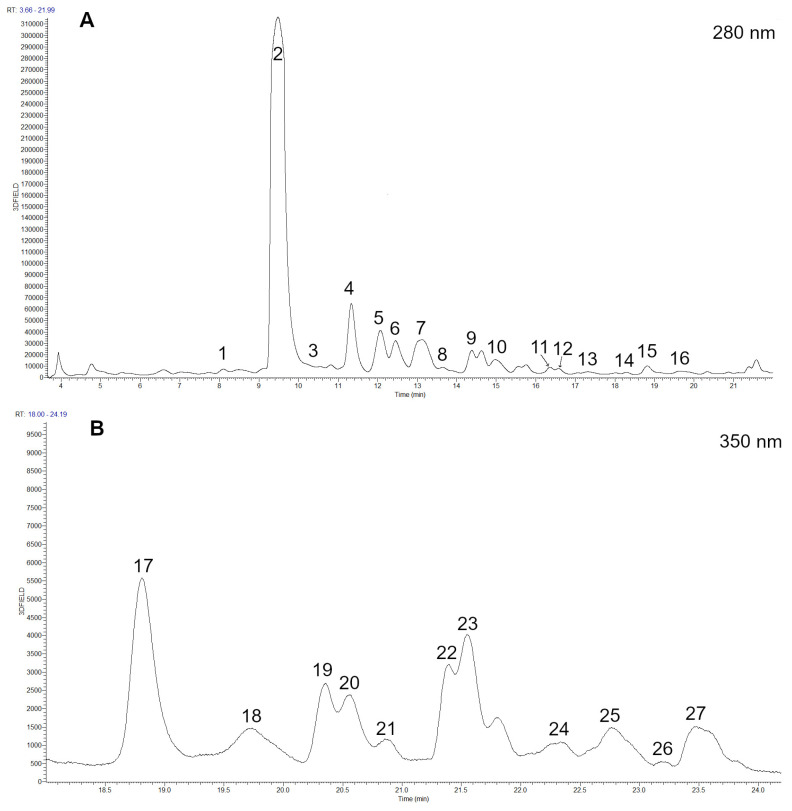
Chromatogram showing the phenolic compounds of medlar extract recorded at (**A**) 280 nm and (**B**) 350 nm. Peak numbers are described in Appendix A.

**Figure 2 foods-12-03077-f002:**
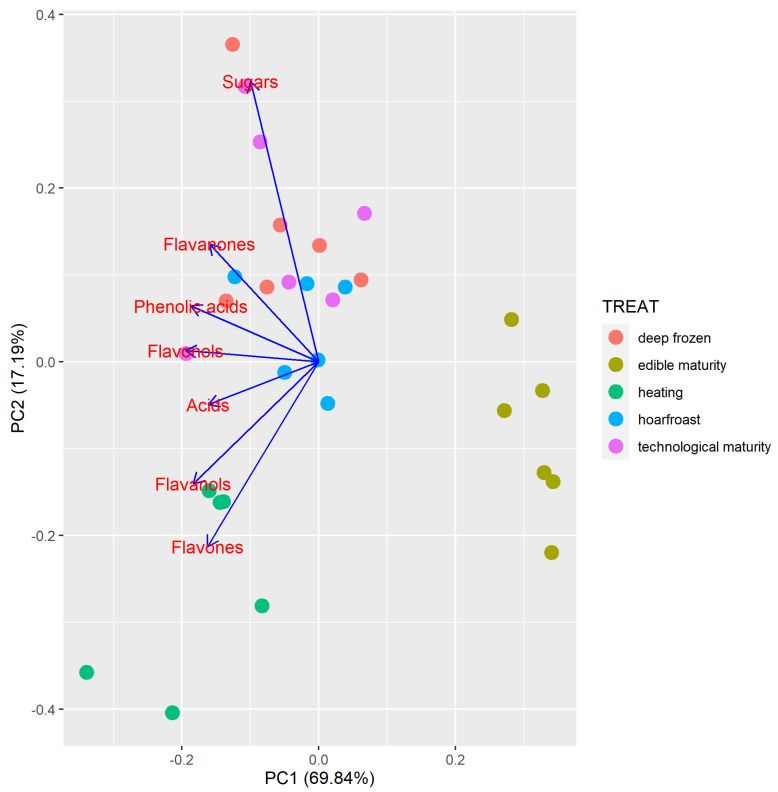
Principal component analysis for medlar fruits treated in different ways (technological maturity, edible maturity, hoarfrost, heating, deep freezing).

**Table 1 foods-12-03077-t001:** Content of sugars and organic acids (g/kg FW or mg/kg FW for shikimic and fumaric acids) in fresh medlar fruit at technological maturity. Means ± standard errors (SE) are presented (*n* = 6).

Compound	Content
Sucrose	4.07 ± 0.29
Glucose	54.43 ± 0.56
Fructose	59.20 ± 1.25
Sorbitol	9.46 ± 0.13
Total sugars	127.16 ± 1.86
Citric acid	1.06 ± 0.18
Malic acid	8.44 ± 0.61
Quinic acid	8.77 ± 0.35
Shikimic acid	34.67 ± 0.95
Fumaric acid	5.31 ± 0.42
Total acids	18.31 ± 1.14
Ratio sugars/acids	7.04 ± 0.35

**Table 2 foods-12-03077-t002:** Content of sugars (g/kg DW) in medlar fruit after different treatments. Means ± standard errors (SE) are presented (*n* = 6).

Sugar	Technological Maturity	Edible Maturity	Hoarfrost	Heating	Deep Frozen
sucrose	14.67 ± 1.02 a	15.79 ± 0.97 a	15.26 ± 0.69 a	13.52 ± 0.92 a	14.82 ± 0.58 a
glucose	196.29 ± 2.00 b	176.92 ± 3.06 c	193.96 ± 2.63 b	192.23 ± 2.20 b	204.38 ± 3.39 a
fructose	213.46 ± 4.51 a	191.77 ± 3.66 b	206.8 ± 4.21 a	195.10 ± 3.15 b	216.00 ± 3.45 a
sorbitol	34.11 ± 0.46 a	29.35 ± 0.94 b	30.02 ± 0.33 b	31.30 ± 1.05 b	35.00 ± 0.95 a
Total sugars	458.53 ± 6.72 ab	413.83 ± 7.62 d	446.04 ± 6.03 bc	432.15 ± 4.53 cd	470.23 ± 7.97 a

Different letters (a–d) in the rows denote statistically significant differences between treatments by Duncan’s multiple range test (*p* < 0.05).

**Table 3 foods-12-03077-t003:** Content of organic acids (g/kg DW) in medlar fruit after different treatments. Means ± standard errors (SE) are presented (*n* = 6).

Organic Acid	Technological Maturity	Edible Maturity	Hoarfrost	Heating	Deep Frozen
Citric acid	3.81 ± 0.65 b	2.79 ± 0.16 b	5.70 ± 0.09 a	1.37 ± 0.42 c	5.11 ± 0.25 a
Malic acid	30.43 ± 2.19 b	19.58 ± 0.41 c	38.79 ± 1.61 a	38.57 ± 0.99 a	33.35 ± 1.29 b
Quinic acid	31.63 ± 1.26 b	25.64 ± 0.51 c	33.30 ± 0.95 b	42.84 ± 1.11 a	32.46 ± 0.71 b
Shikimic acid	0.125 ± 0.003 b	0.099 ± 0.002 c	0.128 ± 0.004 b	0.214 ± 0.003 a	0.121 ± 0.009 b
Fumaric acid	0.0191 ± 0.0015 b	0.015 ± 0.0018 c	0.015 ± 0.0005 c	0.025 ± 0.0008 a	0.021 ± 0.0007 b
Total acids	66.03 ± 3.63 b	48.14 ± 0.81 c	77.94 ± 2.45 a	83.02 ± 1.41 a	71.07 ± 1.19 b

Different letters (a–c) in the rows denote statistically significant differences between treatments by Duncan’s multiple range test (*p* < 0.05).

**Table 4 foods-12-03077-t004:** Content of phenolic compounds (mg/kg DW) in medlar fruit after different treatments. Means ± standard errors (SE) are presented (*n* = 6).

Phenolic Compound	Technological Maturity	Edible Maturity	Hoarfrost	Heating	Deep Frozen
Protocatehuic acid	7.73 ± 0.67 a	0.31 ± 0.10 b	6.68 ± 0.54 a	8.08 ± 0.35 a	7.80 ± 0.63 a
3-caffeoylquinic acid	1708.32 ± 150.22 a	69.17 ± 22.39 b	1477.14 ± 121.18 a	1787.33 ± 79.04 a	1724.01 ± 140.93 a
4-caffeoylquinic acid	43.01 ± 4.05 ab	2.95 ± 0.86 c	37.24 ± 2.88 b	51.66 ± 3.42 a	45.35 ± 3.40 ab
5-caffeoylquinic acid	71.44 ± 9.46 a	16.44 ± 4.49 b	57.94 ± 4.22 a	76.99 ± 6.77 a	68.14 ± 7.24 a
Caffeic acid hexoside	42.89 ± 4.65 a	6.15 ± 1.26 b	31.46 ± 3.64 a	43.05 ± 4.34 a	40.15 ± 4.20 a
Sinapic acid	2.35 ± 0.21 ab	0.31 ± 0.07 c	1.84 ± 0.20 b	2.83 ± 0.19 a	2.64 ± 0.32 a
3-*p*-coumaroylquinic acid	19.63 ± 1.82 ab	2.63 ± 0.59 c	15.43 ± 1.73 b	23.65 ± 1.63 a	22.04 ± 2.67 a
4-*p*-coumaroylquinic acid	2.77 ± 0.42 b	0.88 ± 0.15 c	3.12 ± 0.32 b	5.96 ± 0.76 a	3.27 ± 0.54 b
5-*p*-coumaroylquinic acid 1	5.05 ± 0.86 b	1.73 ± 0.37 c	4.76 ± 0.72 b	11.13 ± 1.32 a	6.03 ± 1.10 b
5-*p*-coumaroylquinic acid 2	3.64 ± 0.73 b	0.36 ± 0.15 c	3.34 ± 0.55 b	9.64 ± 1.36 a	4.27 ± 0.88 b
3-feruloylquinic acid 1	3.84 ± 0.36 ab	0.26 ± 0.07 c	3.32 ± 0.25 b	4.61 ± 0.30 a	4.04 ± 0.30 ab
3-feruloylquinic acid 2	1.92 ± 0.18 b	0.33 ± 0.07 c	2.09 ± 0.29 b	2.94 ± 0.23 a	2.11 ± 0.21 b
Ellagic acid	8.85 ± 1.26 a	0.69 ± 0.13 c	3.92 ± 0.78 b	7.38 ± 0.92 a	6.95 ± 1.40 a
Total phenolic acids	1921.49 ± 170.03 a	102.28 ± 27.65 b	1648.33 ± 135.12 a	2035.32 ± 94.13 a	1936.86 ± 156.50 a
Catechin	2.58 ± 0.23 ab	0.34 ± 0.07 c	2.02 ± 0.22 b	3.10 ± 0.21 a	2.89 ± 0.35 a
Epicatechin	139.15 ± 12.99 b	24.25 ± 5.63 c	151.24 ± 21.48 b	212.86 ± 16.97 a	152.28 ± 15.22 b
Procyanidin dimer 1	306.71 ± 28.94 ab	21.1 ± 6.19 c	265.54 ± 20.59 b	368.42 ± 24.42 a	323.43 ± 24.25 ab
Procyanidin trimer 1	149.95 ± 95 a	19.46 ± 6.66 b	136.05 ± 11.98 a	170.09 ± 12.54 a	162.14 ± 13.14 a
Procyanidin trimer 2	110.84 ± 11.89 b	62.93 ± 11.32 c	99.10 ± 7.46 b	156.50 ± 14.41 a	120.57 ± 9.71 b
Procyanidin trimer 3	46.05 ± 8.31 b	19.92 ± 6.99 b	46.53 ± 8.51 b	367.82 ± 74.86 a	59.42 ± 12.84 b
Procyanidin trimer 4	61.33 ± 6.25 b	5.11 ± 2.08 c	50.63 ± 6.66 bc	250.56 ± 36.32 a	74.75 ± 7.17 b
Procyanidin tetramer 1	332.59 ± 53.03 b	69.05 ± 15.60 c	359.95 ± 45.12 b	637.39 ± 63.17 a	352.16 ± 40.59 b
Procyanidin tetramer 2	190.18 ± 29.07 b	60.80 ± 10.63 c	213.92 ± 22.16 b	408.69 ± 52.08 a	224.52 ± 37.64 b
Total flavanols	1353.02 ± 164.81 b	283.00 ± 51.38 c	1325.03 ± 134.37 b	2575.49 ± 217.74 a	1472.20 ± 137.90 b
Apigenin hydroxyhexoside	5.33 ± 1.13 b	0.85 ± 0.26 c	4.87 ± 0.72 b	14.61 ± 1.55 a	5.34 ± 1.15 b
Total flavones	5.33 ± 1.13 b	0.85 ± 0.26 c	4.87 ± 0.72 b	14.61 ± 1.55 a	5.34 ± 1.15 b
Naringenin hexoside 1	3.49 ± 0.37 a	0.14 ± 0.02 c	2.59 ± 0.21 b	2.32 ± 0.33 b	2.43 ± 0.28 b
Naringenin hexoside 2	1.15 ± 0.287 a	0.15 ± 0.033 b	0.79 ± 0.045 a	0.91 ± 0.081 a	0.83 ± 0.034 a
Naringenin hexoside 3	1.29 ± 0.36 a	0.56 ± 0.11 b	0.98 ± 0.08 ab	1.21 ± 0.16 a	0.85 ± 0.17 ab
Total flavanones	5.94 ± 0.99 a	0.86 ± 0.16 c	4.36 ± 0.32 ab	4.45 ± 0.52 ab	4.12 ± 0.43 b
Quercetin-3-galactoside	2.82 ± 0.41 a	0.57 ± 0.12 b	2.18 ± 0.12 a	2.43 ± 0.19 a	2.24 ± 0.22 a
Quercetin-3-glucoside	2.60 ± 0.31 a	0.50 ± 0.11 b	2.07 ± 0.15 a	2.72 ± 0.22 a	2.42 ± 0.28 a
Quercetin-3-rutinoside	2.01 ± 0.50 a	0.27 ± 0.05 b	1.37 ± 0.07 a	1.59 ± 0.14 a	1.45 ± 0.05 a
Quercetin-pentoside	0.61 ± 0.060 a	0.11 ± 0.014 b	0.51 ± 0.040 a	0.59 ± 0.068 a	0.49 ± 0.041 a
Quercetin dirhamnoside	1.74 ± 0.18 a	0.07 ± 0.01 c	1.29 ± 0.10 b	1.16 ± 0.16 b	1.21 ± 0.14 b
Quercetin-rhamnosyl hexoside	3.64 ± 0.35 a	0.69 ± 0.08 b	3.08 ± 0.24 a	3.50 ± 0.40 a	2.91 ± 0.24 a
Laricitrin rhamnoside	2.90 ± 0.31 a	0.12 ± 0.02 c	2.16 ± 0.17 b	1.93 ± 0.27 b	2.02 ± 0.23 b
Kaempferol hexoside 1	0.11 ± 0.013 a	0.02 ± 0.004 b	0.08 ± 0.006 a	0.11 ± 0.009 a	0.10 ± 0.011 a
Kaempferol hexoside 2	1.16 ± 0.15 a	0.43 ± 0.05 b	1.04 ± 0.12 a	1.15 ± 0.18 a	1.16 ± 0.09 a
Dihydrokaempferol hexoside	41.26 ± 4.97 ab	8.36 ± 1.36 c	36.40 ± 3.67 b	51.23 ± 4.28 a	43.51 ± 3.94 ab
Isorhamnetin hexoside 1	0.52 ± 0.131 a	0.07 ± 0.015 b	0.36 ± 0.020 a	0.41 ± 0.037 a	0.38 ± 0.015 a
Isorhamnoside hexoside 2	1.29 ± 0.24 b	0.89 ± 0.17 b	1.07 ± 0.18 b	2.30 ± 0.17 a	1.05 ± 0.19 b
Total flavonols	60.71 ± 6.44 ab	12.15 ± 1.86 c	51.65 ± 4.59 b	69.17 ± 5.66 a	58.99 ± 5.23 ab
Total analyzed phenolics	3346.51 ± 326.19 b	399.16 ± 77.36 c	3034.26 ± 272.34 b	4699.06 ± 284.51 a	3477.52 ± 287.76 b

Different letters (a–c) in the rows denote statistically significant differences between treatments by Duncan’s multiple range test (*p* < 0.05).

## Data Availability

Data are available from the corresponding author on reasonable request.

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
