# Peer review of "Changes in the Fruit Quality Parameters of Medlar Fruit (Mespilus germanica L.) after Heat Treatment, Storage, Freezing or Hoarfrost"

_foods, 2023, doi:10.3390/foods12163077_

Round 1

Reviewer 1 Report

The article is relatively good. Title must be corrected. Some sentences are inappropriate. Introduction section has been prepared unsuitable. There is no any background around the topic. Much volume of this section belongs to food and medicine properties of medlar. In my opinion, this section must be re-written. Some words have been written using all capital letters. Authors have used “We or our” frequently. Reasons for changes in the content of primary and secondary metabolites must be explained in Discussion section. Other comments have been presented in the text of the article.

Author Response

We would like to thank the reviewer for the detailed review of the article and all the positive comments in the pdf document. We have accepted and included all the reviewer's comments in the article. All changes are highlighted in yellow. We are very grateful for all their precise corrections, which have improved the quality of the manuscript.

Reviewer 2 Report

Medlar fruits of our local genotype 'Domača nešplja' have not been chemically analysed before, in this study, Maja Mikulic-Petkovsek and colleagues investigated which processing method could obtain the best chemical quality of the fruits. The results of the study should provide original and useful information to the scientific community, food-manufacturing industry, private institutions and common consumers. However, we have some concerns about the manuscript, below are the comments.

General comments:

1. The abstract section lacks a background, please provide a brief background in this section.

2. line 13: “and in organic acids, malic acid (8.44 g/kg FW) and quinic acid (8.77 g/kg FW) were.” The expression needs to be modified.

3. The logic of the Introduction section needs to be revised. For example, the fourth and fifth paragraphs can be summarized as one paragraph. In general, this study should introduce the nutritional components firstly, and then introduce the bioactive compounds, however, the L54 and L68 all mentioned phenolic compoundsthe logic is chaotic, please modify it.

4. L99-L108 this section belonging to sample pre-treatment, and not belonging to Plant material section. Please modify.

5. L124 “4mM” Space required between numbers and units.

6. Each of phenolic compounds required the standard curves, please add them to the method.

7. L190-191, “For technologically mature medlar, we also reported results based on fresh weight (FW).” does not belong to statistical analysis section. Please modify.

8. “3.1 Content of primary and secondary metabolites in fresh technologically mature medlar fruits” why would it stand out primary and secondary metabolites? I think it would be clearer to directly state the specific components measured as the title.

9. In my opinion, components 9 and 10, 11 and 12, 19 and 20, 22 and 23 have not been completely separated. How did you conduct qualitative and quantitative analysis?

10. The Figure 2. is blurry, please increase the pixels.

11. Why is 25 missing? Please proofread the accuracy of the references in the entire text.

Quality of English Language need further improvment.

Author Response

General comments:

  1. The abstract section lacks a background, please provide a brief background in this section.

As the reviewer suggested, we have added the objective of the study and slightly changed the text to make it more coherent.

  1. line 13: “and in organic acids, malic acid (8.44 g/kg FW) and quinic acid (8.77 g/kg FW) were.” The expression needs to be modified.

The sentence has been changed.

  1. The logic of the Introduction section needs to be revised. For example, the fourth and fifth paragraphs can be summarized as one paragraph. In general, this study should introduce the nutritional components firstly, and then introduce the bioactive compounds, however, the L54 and L68 all mentioned phenolic compoundsthe logic is chaotic, please modify it.

We thank the reviewer for their helpful comment.WThe text in the INTRODUCTION chapter has been modified.

  1. L99-L108 this section belonging to sample pre-treatment, and not belonging to Plant material section. Please modify.

According to the reviewer’s comment, we added a new title for the sample preparation.

  1. L124 “4mM” Space required between numbers and units.

It has been modified.

  1. Each of phenolic compounds required the standard curves, please add them to the method.

All standard curves for phenolic compounds have been added in the Materials and Methods.

  1. L190-191, “For technologically mature medlar, we also reported results based on fresh weight (FW).” does not belong to statistical analysis section. Please modify.

We have modified the text.

  1. 3.1 Content of primary and secondary metabolites in fresh technologically mature medlar fruits” why would it stand out primary and secondary metabolites? I think it would be clearer to directly state the specific components measured as the title.

Thank you for the comment. the title has been changed.

  1. In my opinion, components 9 and 10, 11 and 12, 19 and 20, 22 and 23 have not been completely separated. How did you conduct qualitative and quantitative analysis?

For phenolic compounds whose peaks were not well separated, we manually split the peaks. Based on the results of the mass spectrum, we knew exactly the ratio in which certain substances were present in the peak, which we calculated from the peak area obtained. In this way, the individual phenolic substances were qualified and quantified using a suitable standard.

  1. The Figure 2. is blurry, please increase the pixels.

The picture has been improved and the resolution is ameliorated.

  1. Why is 25 missing? Please proofread the accuracy of the references in the entire text.

Thank you for the comment. All references were proofread throughout the entire manuscript.

Comments on the Quality of English Language

Quality of English Language need further improvement.

Extensive English proofreading was done by a native speaker.

We have accepted all the reviewer's comments and included them in the text of the article. We are very grateful to the reviewer for all their precise corrections and comments, which have improved the quality of the manuscript.

Round 2

Reviewer 2 Report

Authors have made a better revison and it is a very complete manuscript, which shows A LOT of data and I believe that it deserves to be published in a high impact factor journal like Foods.